# Current Advantages in the Application of Microencapsulation in Functional Bread Development

**DOI:** 10.3390/foods12010096

**Published:** 2022-12-24

**Authors:** Roberta Tolve, Federico Bianchi, Elisabetta Lomuscio, Lucia Sportiello, Barbara Simonato

**Affiliations:** 1Department of Biotechnology, University of Verona, Strada Le Grazie 15, 37134 Verona, Italy; 2School of Agricultural, Forestry, Food and Environmental Sciences (SAFE), University of Basilicata, 85100 Potenza, Italy

**Keywords:** microencapsulation, bread, bioactive compounds, functional food, shelf-life, sensory properties

## Abstract

Bread is one of the most widely embraced food products and is highly accepted by consumers. Despite being rich in complex carbohydrates (i.e., starch), bread is generally poor in other micro- and macronutrients. Rising consumer demand for healthier food has resulted in the growth of studies focused on bread fortification with bioactive ingredients (i.e., vitamins, prebiotics, and vegetable extracts). However, the baking process leads to the reduction (or even lessening) of the added substance. In addition, the direct inclusion of bioactive compounds and additives in bread has other limitations, such as adverse effects on sensory characteristics and undesirable interaction with other food ingredients. Encapsulation allows for overcoming these drawbacks and at the same time improves the overall quality and shelf-life of bread by controlling the release, protection, and uniform distribution of these compounds. In the last ten years, several studies have shown that including micro/nano-encapsulated bioactive substances instead of free compounds allows for the enrichment or fortification of bread, which can be achieved without negatively impacting its physicochemical and textural properties. This review aims to identify and highlight useful applications in the production of new functional bread through encapsulation technology, summarizing the heath benefit and the effect of microcapsule inclusion in dough and bread from a technological and sensory point of view.

## 1. Introduction

Bread is a staple cereal product with a high consumer acceptance worldwide. Globally, three types of bread are produced: wheat, gluten-free, and mixed bread (obtained from a blend of different flours) [1]. However, most bread is made from refined flour, which is obtained through the removal of bran and germ, and thus is poor in notable nutrients, such as vitamins, minerals, and antioxidants [2]. In addition to the milling and refining process, dough usually undergoes a thermal process during bread production [3]. Baking is essential for digestibility, palatability, and the development of physical and nutritional bread characteristics; at the same time, however, baking leads to the reduction, or even the destruction, of some bioactive compounds [4]. In this perspective, common bread cannot meet the growing nutritional and healthy needs of consumers in contrast to enriched or fortified bread that is obtained by the addition of bioactive compounds or by the inclusion of a lost component to the formulation [3,5]. To date, various attempts to improve the nutritional characteristics of bread have been made: adding crude vegetable or fruit extracts, or even adding phenolic and dietary fiber-rich byproducts to increase antioxidant content and modulate starch digestion. Even though the enrichment of bread in terms of antioxidants and dietary fibers and a reduction of carbohydrates intake have been accomplished, several problems have sometimes stood out, such as the reduction of sensory acceptability, lower volume, lower shelf-life, and higher firmness of fortified bread [6,7]. The direct addition of bioactive compounds has also been performed, but unpleasant flavor and taste may limit this application. Moreover, thermal instability, high volatility, low bioavailability, and bioaccessibility hamper bread fortification [8]. These and other obstacles can be overcome through the encapsulation of bioactive compounds in protective matrices before their use for food enrichment/fortification. The encapsulation process could be defined as the process by which substances in the solid, liquid, or gaseous state are surrounded by a coating or embedded in a matrix given different size particles [9]. Many researchers have applied encapsulation technology for the development of functional bread with encapsulated probiotics, enzymes, vitamins, polyphenols, and omega-3 fatty acids [10,11,12,13,14,15]. However, the addition of the encapsulated bioactive compounds to the bread requires the evaluation of different aspects, such as the health benefits, technological, rheological, and sensory properties, and product shelf-life; in the case of probiotics, it is mandatory to perform an evaluation of the survival in the gastrointestinal tract. All of these aspects must be carefully assessed and balanced as they determine the functionality and boost consumer acceptance of the final products [16]. The present review aims to explore the usefulness of the encapsulation technique, which is used to overcome the main problems related to the fortification of bread with bioactive compounds; the investigation is divided according to the nature of the added substance. Through a thorough scrutiny of the most pertinent and recent articles, this review highlights the available information on the effect of the inclusion of encapsulated bioactive compounds on bread’s technological and sensory properties; this review also discusses the results using a critical approach. At the same time, the review points out the main shortcomings and the lakes of information in this research field related to specific bioactive compounds, which should be investigated.

### Microencapsulation Process and Techniques

Microencapsulation technology was developed about 70 years ago for the pharmaceutical sector. Since then, its application has extended to the agri-food, biotechnological, textiles, and cosmetics fields. In the food sector, the encapsulated substance, the “core”, is the bioactive compound, and the coating, referred to as the “shell” or “wall”, is the polymeric matrix. The coating material should be of food grade, be able to form a barrier between the active agent and its surrounding environment, be tasteless and flavorless, be impermeable, and have the ability to release the core at a specific time and site upon the specific environmental factor [17]. The choice of the coating material depends on several factors, such as the core characteristics, technique applied, compatibility with the food application, and impact on the sensory properties of the final product. The coating material can be polysaccharides, proteins, lipids, or gums that are individually used or used in combination, with or without using surfactants and emulsifiers as additives. Based on the size, the capsules can be classified as micro (1–800 μm) or nano (10–1000 nm) [18]. Several morphologies can be obtained, but the three major ones are mononuclear, polynuclear, or matrix types. The distinctive shape of these structures is related to the nature of the core and wall materials as well as the process technologies selected for their production [19]. The physicochemical properties of microcapsules depend on the chemical nature of the core materials, the physical structure and chemical properties of the wall materials, the interaction between the core and the wall, and the working conditions of the process. Specifically, the encapsulation process is considered a viable option in the food sector for improving storage stability, masking unpleasant flavors or tastes, obtaining target delivery, increasing their solubility, or avoiding adverse ingredient interaction [20]. In addition to the encapsulation yield, which is a process parameter determined as the ratio between the microcapsules collected and the initial quantity of solids contained in the suspension/emulsion/solution, other parameters are evaluated, such as the encapsulation efficiency and the loading capacity. The encapsulation efficiency is the ratio between the bioactive compound in the microcapsules divided by the total amount of bioactive compound added. The loading capacity is instead the amount of bioactive compound loaded per unit weight of the microcapsule, indicating the percentage of the nanoparticle mass that is due to the encapsulated compounds [21].

Microencapsulation technology can be categorized into physical, chemical, and physicochemical methods. In the physical method, microencapsulation is based on physical and mechanical principles. The microcapsule formation depends on the solid–liquid phase transition under heating or the solubility reduction due to the evaporation of the solvent. The physical methods of formation include spray drying, spray-chilling, freeze drying, vacuum drying, fluid bed coating, extrusion, stationary nozzle coextrusion, centrifugal coextrusion, vibrational nozzle, air, and centrifugal extrusion. The chemical methods involve chemical reactions, in which the monomers with small molecules polymerize to form the polymer shell as in interfacial and in situ polymerization. Conversely, in physicochemical processes, the pre-dissolved shell materials precipitate from the solution following the variation in temperature, pH value, or electrolyte concentration. The shell gradually deposits on the core material, forming the microcapsules by means of ionotropic gelation, solvent evaporation, layer-by-layer adsorption, polyelectrolyte complexation, coacervation, and phase separation [18]. Among them, the most common bioactive compound encapsulation technologies applied to the food sector are spray drying, freeze drying, coacervation, fluidized bed coating, and extrusion (Table 1).

## 2. Bioactive Compounds Microencapsulation to Enrich and Fortify Bread

Several authors have focused their efforts on bread development with encapsulated bioactive compounds. Frequently, the inclusion of encapsulated bioactive compounds was carried out to enhance the nutritional value of bread with substances that could exert a positive effect on human health. In this regard, consider the action of vitamin D on the immune system, polyphenols on antioxidant and radical scavenging activities, and probiotics on the well-being of the gastrointestinal tract. Other times, the inclusion of the encapsulated substances in bread had mostly technological and sensorial purposes, as for the case of enzymes.

### 2.1. Vitamins and Minerals

Vitamins and minerals are essential micronutrients that assist the growth, development, and metabolic processes of human beings [22]. Micronutrient intakes are minimal for physiological functions to maintain health. However, their deficiencies affect up to two billion people and cause three million childhood mortalities each year [23]. Typically, as a strategy in the management of micronutrient deficiency, the bioactive compounds are directly mixed into the food matrix. However, in this way, nutrients are easily subjected to high temperature, oxygen, and humidity during the storage or the cooking processes; thus, they are prone to chemical degradation, resulting in the loss of nutritional properties. Microencapsulation has been used as a valid technique for preserving the biological activity and stability of vitamins and minerals (Table 2). Among the vitamins, the one that is most often deficient worldwide is vitamin D. Vitamin D is a fat-soluble precursor of calcitriol, an active form of vitamin D, which is critical to the regulation of phosphorus and calcium homeostasis. The deficiencies of vitamin D can be countered by using food fortification strategies [15]. However, the challenge of including vitamin D in food is due to its high sensitivity to oxidation when exposed to heat, light, moisture, or oxygen [21]. To overcome this drawback, Zhu et al. [24] studied the microencapsulation of vitamin D with egg white and its application for bread fortification. The authors applied ultrasonication, an emerging technology that uses soundwaves above 20 kHz. Specifically, the authors layered vitamins on the surface of an aqueous egg solution (used as wall material) and sonicated the solution for 1 min at 160 W. The encapsulated compounds’ protection from photodegradation required a further coating with green tea catechin/iron. The encapsulated form of the vitamin proved to be more resistant to light and heat than the non-encapsulated type. In addition, the authors reported that the inclusion of free vitamin D in the dough tended to coalesce, merge, and form oily regions. Indeed, egg white microcapsules were proved to protect vitamin D from mechanical stresses. Bread fortified with microcapsules showed a higher vitamin D recovery (81.3%) than that with free vitamin D, indicating that the microcapsules embedded into food matrices were thermostable and that the vitamin D was protected from degradation to a certain extent. Accordingly, Constantino et al. [25] reported a rate of recovery of 88.65% for vitamin D microencapsulated in complexes and formed by amaranth protein isolates and lactoferrin after the baking process compared with the 64.63% obtained for the non-encapsulated vitamin. In both studies reported [24,25], the in vitro gastrointestinal digestion process of bread was performed, and it was reported that the greatest amount of vitamin D was released from the microcapsules in the intestinal phase. Specifically, Costantino et al. [25] reported that 22.98% of the vitamin was released in the first 5 min and 82.96% was released by 180 min for the encapsulated sample. Similarly, vitamin D in an egg white designed microcapsule allowed 67% of vitamin D to reach the intestine in the active form. whereas the free sample allowed only 32% of the ingested vitamin to be absorbed in the intestine due to the degradation in the stomach [24]. In addition to vitamin D, microencapsulation has also been investigated to improve the thermal stability of some other vitamins during bread baking. Folic acid, for example, has been proven useful for reducing the incidence of neural tube defects, but there are concerns that it could mask vitamin B12 deficiency. A reduced form of folate, L-5-methyltetrahydrofolic acid (L-5-MTHF), is nowadays used for food fortification. However, it is susceptible to oxidation, leading to important losses during bread baking. Liu et al. [26] reported an important increase in L-5-MTHF stability during bread baking upon microencapsulation by spray dryer with modified starch as a wall material. In addition, the co-encapsulation of L-5-MTHF and ascorbate promoted the storage stability of bread and recovery of L-5-MTHF after bread baking as compared with free L-5-MTHF. The L-5-MTHF recovery rate also remained high after the scale-up process, passing by 97% for the pilot plant compared with 77% for the commercial bakery scale. Similarly, Tomiuk et al. [27] reported that L-5-MTHF encapsulated via a spray dryer using skim milk powder as the wall material remained highly stable (>80%) in white bread. Sodium ascorbate with skim milk powder as shell materials had an even better positive effect on the stability of folic acid. In contrast, Neves et al. [28] reported a higher thermal degradation rate of microencapsulated folic acid compared with its free counterpart when included in French-type bread. The results showed that the free folic acid degradation was completed at 155 °C/30 min, whereas the microencapsulated folic acid was entirely degraded at 100 °C/15 min. The authors reported that the high surface area of capsules and locating folic acid on the surface of particles could be responsible for this behavior.

Regarding minerals, major deficiencies are related to iron and zinc. A third of the population worldwide suffers from zinc deficiency, while more than half have an iron deficiency. Iron is an essential structural component of hemoglobin, myoglobin, and cytochrome-dependent proteins. Iron deficiency, the most common cause of anemia, affects 1.5–2 billion people worldwide. It is associated with diminished work productivity, lower immunity, and impaired cognitive development. Three main reasons for iron deficiency can be identified: inadequate iron intake, compromised bioavailability, and increased iron losses [29]. Food fortification is a dietary strategy that can be used to overcome this nutritional disease. However, food fortification may result in the undesirable color, odor, and taste of foods. Furthermore, iron may interact with food protein and lipids, decreasing its bioavailability. Iron encapsulation can assist with improving sensorial properties in fortified foods [11]. Bryszewska et al. [11] evaluated the bioaccessibility and bioavailability of microencapsulated iron in bread prepared by conventional and sourdough fermentation using the human epithelial adenocarcinoma cell line Caco-2. Besides the control, four breads fortified with two iron sources (ferrous sulphate and lactate) with or without ascorbic acid were produced. The iron bioaccessibility of the fortified bread after gastrointestinal digestion varied from 35.99 to 99.31%. Specifically, samples obtained with the conventional bread-making process showed the highest iron bioaccessibility. Furthermore, the percentage of iron transport efficiency showed no differences between the types of bread fermentation. In comparison with other samples, the microcapsules with iron and ascorbic acid showed a higher iron transport efficiency (13.78%). This effect is probably due to the antioxidative properties of this vitamin preserving iron in the divalent form, which is more available for transport by enterocytes.

Overall, the studies reported show that the inclusion of microencapsulated vitamins and minerals in bread increases their concentration and bioavailability. However, neither the studies on vitamin D and folic acid nor the one on iron microencapsulation evaluated the microcapsule’s inclusion effects on the rheological, technological, and sensory characteristics of bread. It has only been reported that the use of microencapsulated vitamin D avoids the coalescence, merging, and formation of oily regions in bread. However, it would be necessary to deepen other aspects related to the effects of the inclusion on bread quality and, on account of the bioactives lability, to conduct an in-depth assessment of its stability during the bread shelf-life.

### 2.2. Polyunsaturated Fatty Acids (PUFAs)

Polyunsaturated fatty acids such as omega-3 and omega-6 are highly desirable in bread fortification as they could prevent cardiovascular and inflammatory diseases, cancer, and metabolic syndromes [30]. These fatty acids are very labile from a technological point of view and are susceptible to oxidative deterioration. Bread fortified with PUFAs could develop undesirable fishy flavors, resulting in decreased sensory qualities and food shelf-life. Indeed, the encapsulation of oils can inhibit or delay their oxidation and mask negative flavors and unpleasant tastes [31]. In addition, as reported by several authors, the encapsulation of oils could protect them in the gastrointestinal tract without altering their chemical characteristics [32,33]. Moreover, the addition of encapsulated oils in bread could affect most of the technological and sensory properties, such as bread volume, texture, fatty acids bioavailability, the bread’s oxidative stability, and sensory properties (Table 3).

#### 2.2.1. Bread Oxidative Stability

González et al. [32] studied the fortification of bread with encapsulated chia oil using soy protein as the coating material and assessed the hydroperoxide values up to 14 days in storage. The research showed the protective effect of microencapsulation against oil oxidation. Indeed, bread with microencapsulated chia oil presented three-fold fewer hydroperoxides compared with bread with free oil. Moreover, the protein–polysaccharide matrix (soy protein isolate–maltodextrin–pectin) used as wall material for Himalayan walnut oil was able to prevent the bread oxidation in terms of peroxide value, anisidine value, and acid value during 8 days of storage. The soy protein isolate and pectin as wall material were able to delay fatty acids oxidation due to their antioxidant properties. In addition, the scanning electron microscopic analysis showed intact encapsulated oil bodies in the crumbs after baking, thereby demonstrating high omega fatty acid retention [34]. Sodium alginate, used as a coating material for the encapsulation of garlic oil by Narsaiah et al., [35] was reported as an effective barrier against oxygen and was useful for bread fortification. In addition to the coating polymers, the nature of the core had a significant impact on the oxidation of the fortified bread. Regarding this, Sridhar et al. [33] and Kairam et al. [36] demonstrated a synergistic effect between the mixtures of oils selected as the core material (garlic oil combined with fish or flaxseed oil) in reducing oil oxidation throughout the storage period. In these studies, the secondary oxidation products during a week of storage were lower in bread fortified with the oils blend (0.32 µmol of malonaldehyde/g at day 7 with fish and garlic oil as compared with 0.48 µmol of malonaldehyde/g at day 7 for bread fortified with only fish oil) [33]. Similar results were reported also by Kairam et al. [36].

#### 2.2.2. Bread Textural Properties 

Ojagh and Hasani [37] fortified bread with liposome-encapsulated fish oil and observed an increase in volume by 5% of the microcapsules addition. These results are referring to the characteristics of the coating polymers and, specifically, to the surface-active properties of lecithin used as an emulsifier within the liposomal system, which would have determined the improvement in gas retention, bread volume, and dough stability. Akhtar et al. [34] and Takeungwongtrakul et al. [38] suggested that the hydrophilic nature of the colloidal molecules used as a coating material for PUFAs microencapsulation could influence bread volume. Specifically, functional bread developed with the inclusion of Himalayan walnut oil encapsulated into the soy protein isolate–maltodextrin–pectin complex in different percentages and was characterized by a higher specific volume compared with the control bread [34]. In this case, the volume increase could be ascribed to the formation of the polysaccharide–soy proteins complex in the functional dough that could hinder cell coalescing, promote the alveolar formation, and thus increase bread volume. This thesis was corroborated by Takeungwongtrakul et al. [38], whose study investigated the incorporation of microencapsulated shrimp oil, another PUFAs source, in bread samples. The authors reported that whey protein concentrate selected as wall material might strengthen the bread structure by interacting with the gluten network and promote hold gas retention more efficiently, thus increasing the final volume. In contrast, Costa de Conto et al. [39] fortified bread with an increasing amount of commercial omega-3 and rosemary extracts microcapsules and observed a decrease in the specific volume that was inversely correlated with the microcapsules addition. The authors explained that microcapsules incorporation could dilute the gluten and interfere with the retention of gases, resulting in less volume. Similarly, bread with chia oil microencapsulated in soy protein isolate and maltodextrin showed a lower specific volume than bread with free oil. However, no significant differences between the bread fortified with microencapsulated oil and the control sample (without the oil) were reported. The non-encapsulated oil plasticizes and lubricates the gluten polymers of the dough, increasing dough rise and loaf volume. In addition, the inclusion of microencapsulated chia oil in bread determines a lower cell average area than other samples, increasing crumb uniformity [32]. The variation in the specific volume of loaves is linked to the bread’s textural properties, hardness, cohesiveness, and springiness, which are qualitative parameters that determine the texture of the bread. In general, it is desirable to have low hardness values in bread. That is what has been obtained by Akhatar et al. [34], which fortified bread with encapsulated Himalayan walnut oil obtained through a complex coacervation technique and observed a crumb hardness decrease (by up to 65%). The hydrophilic nature of pectin used as a wall material with soy protein isolate and maltodextrin increased the water holding capacity and moisture in functional samples, thus reducing their hardness. A similar trend was observed by Takeungwongtrakul et al. [38] using whey protein concentrate as the coating material. The hardness reduction was also detected during the storage of bread fortified with microencapsulated garlic oil [35]. The hardness reduction may be due to the hydrophilic nature of the alginate used as shell material for garlic oil, which helped the retention of moisture. Similarly, the hardness reduction in bread fortified with liposome encapsulated-fish oil has been reported [37]. This would be due to the presence of lecithin and glycerol emulsifiers in the liposomal encapsulated structure. Contrarily, another study included the subsequent addition (from 0% to 5%) of a commercial microencapsulated omega-3 fatty acids and rosemary extract in bread, which caused a linear hardness increase [39]. However, to the best of our knowledge, the wall material(s) used in this work have not been reported.

#### 2.2.3. Bread Sensory Properties

Generally, the modification of the technological and rheological characteristics of bread fortified with microencapsulated bioactive compounds proceeds side-by-side with the variation of the sensorial aspects. The addition of fish oil nano-liposomal capsules in bread samples improved the crumb color, aroma, taste, and overall acceptability, resembling the values of the control sample [37]. Similarly, garlic oil microcapsules in bread enhanced all of the sensory parameters compared with bread with free garlic oil after a week of storage, presumably due to the lower oxidation rate of garlic oil within the microcapsules [35]. Similarly, the improvement of the sensorial characteristics and the bread acceptability was also reported during 7 days of storage with samples fortified with a blend of oils [35,36]. However, these positive effects on the sensory characteristics of bread fortified with microencapsulated source of PUFAs appears to be dependent on the concentration of microcapsules used.

**Table 3 foods-12-00096-t003:** Encapsulation of fatty acids sources for bread fortification.

Core	Wall	Technique	Keys Finding(s) and Recommendation						
			EE	Volume	In vitro Bioavailability	Oxidative Stability	Texture	Sensory Analysis	Ref.
Flaxseed oil	Yeast cell and Oat β-glucans	Freeze drying	n.d.	↓ MCs bread vs. control and free oil bread	n.d.	↓ PV for 7 days storage in yeast cells MCs	↑ Firmness on day 1 and during 7 days of storage	↓ Softness, hardness ↑ flavor MCs vs. free oil ≈ overall acceptability	[40]
Omega-3 and Rosemary extract	n.d.	n.d.	n.d.	↓ MCs vs. control bread	n.d.	n.d.	↑ Firmness	↓/≈ Appearance, aroma, and overall acceptability	[39]
Chia oil	Soy proteins	Freeze drying	n.d.	↓ MCs vs. free oil bread ≈MCs vs. control bread	AO: 94.83%	3-fold fewer HPV in MCs vs. free oil	≈ Firmness, springiness, cohesiveness, chewiness (MCs bread vs. control bread during 14 days of storage)	↑ Overall acceptability (MCs vs. control bread) ↓odor (MCs vs. free oil bread)	[32]
Fish oil	Glycerol	Nano-liposomes	90.12 %	↑ 5% MCs bread at day 0 and after 3 days of storage	n.d.	↓PV ↓TBARS during 25 days of storage	↓ Hardness and gumminess ≈ Springiness, cohesiveness (at day 0 and after 3 days of storage)	↑ Appearance, crumb, aroma, taste, overall acceptability (MCs vs free oil bread.	[37]
Fish oil	Chitosan and Hi-Cap100	Freeze drying	Up to 79.37% (CS:Hi-Cap100 1:9)	n.d.	n.d.	n.d.	↑ Firmness (up to 2.5% MCs)	↓ Appearance, taste, texture, aroma, crumb and overall acceptability (up to 2.5% MCs)	[41]
Flaxseed and garlic oil	Sodium alginate	n.d.	n.d.	n.d.	n.d.	TBARS: Control (without oil) < GO < FL-GO < FL	↑ Hardness ≈ cohesiveness and springiness over 7 days of storage time	↑ Flavor and color FL-GO vs. GO and FL	[36]
Garlic oil	Calcium alginate	Nanoemulsions	n.d.	n.d.	n.d.	↓ TBARS GO vs. free oil	↓ Hardness ≈ cohesiveness and springiness up to 7 days of storage	↑ Appearance, color, texture, aroma, favor, overall acceptability GO MCs vs. free oil during storage	[35]
Fish and garlic oil	Soya lecithin	Microemulsions	n.d.	n.d.	86.89%(FO) 61.36% (GO) 70.90% (FO-GO)	TBARS: Control (without oil) < GO< FO-GO< FO	≈ Cohesiveness and springiness during storage	↑ Flavor, aroma, and overall acceptability FO-GO vs. FO throughout 7 days	[33]
Shrimp oil	WPI-C sodium caseinate glucose	Spray drying	n.d.	↑ 1–5% MCs vs. control	n.d.	n.d.	↓ Hardness MCs vs. control ≈ springiness and cohesiveness during 3 days of storage	↓ Odor and overall acceptability MCs vs. control during storage	[38]
Himalayan walnut oil	SPI + MD + Pectine	Freeze drying	n.d.	↑ MCs bread vs. control bread	n.d.	↓ PV, AnV and AV	↓ Hardness	n.d.	[34]

Encapsulation efficiency [EE]; Microcapsules [MCs]; fish oil [FO]; garlic oil [GO]; maltodextrin [MD]; whey protein isolate-concentrate [WPI-C]; modified starch from waxy maize [Hi-Cap 100]; soy protein isolate [SPI]; peroxide value [PV]; hydroperoxide values [HPV]; amount of available oil after digestion [%AO]; anisidine value [AnV]; acid value [AV] ≈ no effect/no significative effect; ↑ higher quantity/effect; ↓ lower quantity/effect; n.d.: no data.

Hasani et al. [41] reported how the microencapsulation of fish oil could balance the fishy flavor and odor of fortified bread, highlighting a reduction in the taste, aroma, and overall acceptability of bread fortified with 5% of microencapsulated fish oil. Instead, there were no significant differences between the control sample and the one containing 1% microcapsules with respect to appearance, texture, and crumb. Likewise, Takeungwongtrakul et al. [38] reported the need to balance the amount of microcapsules used for bread fortification with oil rich in PUFA. The authors specifically reported that a 5% inclusion of shrimp oil microencapsulated using whey protein concentrate, sodium caseinate and glucose syrup caused a reduction in the odor, appearance, and overall likeness scores of bread.

### 2.3. Phenolic Extracts

Much research has been devoted to the encapsulation of plant extracts to be added into bread to improve the nutritional value, structural properties, and microbial stability of this product (Table 4). Specifically, when considering the incorporation of polyphenols, the encapsulation technique represents a valid solution to overcome their sensibility to high processing temperatures and alkaline pH as well as the possible undesirable increase in astringency and bitterness in the fortified products. Bread antioxidant properties and the content of phenolic compounds have been enhanced using extracts derived from several sources, such as Garcinia cowa fruit [10,42], green tea [43], Saskatoon berry fruit [44,45], and the bark of soybeans, onion husks, and young hawthorn shoots [46].

#### 2.3.1. Bread Phenolic Content and Antioxidant Activity

Bread fortified with microencapsulated Garcinia cowa fruit extract showed higher [-]-hydroxycitric acid (HCA) concentration in spray dryer capsules with maltodextrin as wall material compared with others using whey protein and a combination of the above wall materials [10]. Furthermore, comparing with the results of their subsequent study, freeze-dryer encapsulates incorporated in bread retained a higher HCA due to higher encapsulation efficiency [42]. Similar results were observed by Pasrija et al. [43] when comparing freeze dryer and spray dryer technology with the same wall materials. The higher bread catechin content was assumed to be due to the higher drying rate of maltodextrin and rapid crust formation, which leads to the retention of polyphenols. Moreover, Lachowicz et al. [44] investigated the phenolic content and antioxidant activity of rye bread fortified with microencapsulated Saskatoon berry fruit. Again, the bread with maltodextrin microcapsules showed the highest phenolic content and antioxidant activity, increasing by 91% and 53%, respectively, compared with the control bread. In particular, the microencapsulation process ensured protection from the degradation of anthocyanins and phenolic acids. Furthermore, Czubaszek et al. [46] proposed the microencapsulation of extracts from the bark of soybeans, onion husks, and young hawthorn shoots using inulin and maltodextrin. The data obtained showed that soybean extract represents the richest source of polyphenols. In contrast, the lowest increase in these compounds in bread was obtained with hawthorn. Moreover, an overall higher protective effect of maltodextrin than inulin was observed, except for with the onion husk extract. However, bread samples with microencapsulated onion husk extract and hawthorn bark showed the highest antioxidant activity, by more than sixty times, compared with the control bread. In the study, the bioavailability of the incorporated bioactive compounds performing in vitro digestion was also investigated. Unfortunately, due to the variability of the data collected for polyphenols and antioxidant activity, the authors were not able to identify which of the wall materials used exhibited a greater protective effect on polyphenols.

#### 2.3.2. Bread Technological Parameters

The addition of encapsulated phenolic extract in bread could affect most of its technological and sensory properties, such as bread volume, moisture content, texture, color, and taste. Ezhilarasi et al. [42] observed a lower volume for bread with encapsulated Garcinia cowa extracts compared with the control bread. The authors suggested that the added acid (HCA) could lower the dough pH, liming the yeast growth [47] or extensibility of the dough [48]. However, differences among the diverse experimental formulations were observed. Bread with whey protein isolate microcapsules exhibited a higher volume and softer crumb texture than bread with maltodextrin and a mixture of whey protein isolate and maltodextrin encapsulates when a freeze-drying technique was applied [42]. In contrast, in the second study where spray drying was used as the encapsulation technology, bread with maltodextrin encapsulates resulted in softer crumb texture and notable volume [10]. The results reported in the two studies highlight that there was an effective higher protection of the phenolic extract by the spray drying technique, demonstrate the variations in the encapsulating properties of the coating materials under two different techniques selected, and demonstrate the effects on the bread qualitative characteristics. Similarly, Czubaszek et al. [46] reported a lower volume for bread enriched with inulin or maltodextrin soybeans, onion, and young hawthorn encapsulated extracts. These results underlined that the source of the extract and the coating material used influenced the bread volume. In contrast, Pasrija et al. [43] observed no significant difference in the volume of bread fortified with unencapsulated green tea extract or encapsulated with a spray dryer or freeze dryer, although a lower moisture content in control bread was reported. The authors stated that polyphenols and the presence of the dietary fiber used as wall material (maltodextrin + β-cyclodextrin) may have promoted water retention due to the interaction of carbonyl, amine, or hydroxyl-functional groups with water. Conversely, in Ezhilarasi et al. [10], bread with maltodextrin, whey protein isolate, or their combination microcapsules of Garcinia cowa extract had similar moisture content compared with the control wheat bread. Microcapsules inclusion may affect other important quality parameters, such as the bread color. Lachowicz et al. [45] fortified bread with dried or encapsulated Saskatoon berries with inulin or maltodextrin as coating materials. The free berry powder addition reduced the L* parameter, resulting in darker loaves. However, the reduction was pronounced in the crumb for bread with microcapsules, whereas the effect on the crust was lighter. This means that the crumb of the bread contains more red and less yellow pigment, whereas the crust was less red and yellow. Thus, the addition of the encapsulated extract masked the red color of the fruit used. Similar results on color were observed by Ezhilarasi et al. [42] when wheat bread was enriched with microencapsulated Garcinia cowa fruits compared with bread containing only the aqueous extract. Moreover, Pasrija et al. [43] reported that microencapsulated extract used in the bread exhibited favorable color parameters, which were due to the reduced direct exposure of bread to green tea extract.

#### 2.3.3. Bread Sensory Acceptability

It emerged in studies published by Ezhilarasi et al. [10,42] that the acceptability of fortified bread was strictly dependent on the technology used and the coating polymers. When the microcapsules were obtained by freeze drying, isolated whey proteins ensured a better acceptability of the product. On the other hand, when the technology applied was spray drying, the bread with the highest level of acceptability was the one fortified with Garcinia cowa microencapsulated in maltodextrins. Recently, two studies dealing with the inclusion of Saskatoon berry fruit in rye and wheat bread were published [44,45]. The extract was microencapsulated using maltodextrin and inulin as coating materials, and the freeze-dried microcapsules were added to bread at different concentrations (from 1 to 6%). For both bread types, the researchers emphasized the importance of investigating the level of inclusion. Using a sensory acceptability analysis based on a hedonic scale combined with color analysis, it was found that the inclusion level of the microencapsulated extract should not exceed 3% for both rye and wheat bread. Similarly, Pasrija et al. [43] proposed not to exceed 2% of green tea extract microcapsules in the fortification of bread. However, although at this inclusion level no significant differences were reported in terms of firmness, a slightly lower score of crust and crumb color, taste, and overall sensory acceptability was reported for the fortified samples compared with the control.

### 2.4. Carotenoids 

To date, the research carried out has never investigated the effects of bread fortified with microencapsulated carotenoids on the rheological and sensory characteristics. The main information reported is related to the increase in the bioactive compounds in bread, the controlled release of carotenoids, and the enhancement of the antifungal properties of the fortified bread. Working with pure β-carotene/soybean oil mixture and palm oil as core materials, Rutz et al. [49] tested, for analytical purposes, the use of complex coacervation and ionic gelation methods for preparing microcapsules using chitosan/carboxymethylcellulose as wall materials in the former method and chitosan/sodium tripolyphosphate in the latter. The results showed high encapsulation efficiency for both methods used and higher carotenoid content in microcapsules containing palm oil compared with those with b-carotene/soybean oil mixture. In a subsequent study [50], using palm oil as the core material and using the complex coacervation method, the authors tested the use of different core materials (chitosan/xanthan and chitosan/pectin) and different final steps of encapsulation (lyophilization and atomization). The lyophilized microparticles showed a higher yield and encapsulation efficiency and lower losses in carotenoids than atomized microparticles, but irregular shape and size were observed. When applied to bread, the chitosan/xanthan microparticles led to better releases of carotenoids, and the released compounds were not degraded. Pinilla et al. [51] reported that the use of phosphatidylcholine–oleic acid liposomes for encapsulating garlic extract in wheat bread led to a general enhancement of the antifungal properties of the product because of their high ability to protect the antifungal compounds of garlic from thermal degradation. The encapsulation efficiency was about 80% and the prevention of mold spoilage for wheat bread was documented, with only four and two slides of the bread being moldy at day fifteen of storage, respectively, for samples with free and encapsulated garlic extract; concerning the control samples, they were totally covered by molds in eleven days. Another result of interest is the role performed by the oleic acid in giving the liposomes higher thermal stability compared with pure phosphatidylcholine liposomes.

### 2.5. Probiotic 

Probiotics are beneficial microorganisms that, if present in an adequate amount, has a variety of health benefits, including regulating the microbial flora of the gastrointestinal tract, preventing the growth of pathogenic microorganisms, and controlling the body’s immune responses [52]. Probiotic bacteria can also prevent cardiovascular disease and lower blood cholesterol. Bread is an innovative area in the probiotic food sector and has attracted increasing interest in research. Due to the high temperatures during the bread-baking process, the inclusion of probiotics in bread is challenging. Fortification of bread with free probiotic cells has been little investigated as the microorganism’s survival levels after baking remain low. On the other hand, microencapsulation appears to be a promising solution to overcome these hurdles; however, even now, this has not been extensively explored [53]. Before incorporating microencapsulated probiotics in bread, a characterization of them is essential for evaluating the encapsulation efficiency and resistance to the gastrointestinal environment (Table 5). In most of the research carried out, the encapsulated probiotics are evaluated by undergoing strong acidic treatment and high temperature to explore the resistance of the microcapsule to simulate the acidic environment of the stomach, or baking condition, respectively. Due to the intrinsic different resistance of the diverse probiotics, specific microencapsulation process optimization must be evaluated from time to time. Proof that there is not a valid encapsulation method to be indiscriminately used for all microorganisms is largely reported in the literature [14,54,55,56,57,58,59].

#### Bread Containing Probiotics Characteristics

The inclusion of microencapsulated probiotic cells into bread, as well as the addition of any other ingredient, could lead to a variation in the technological and sensory as well as nutritional characteristics of the product. Surely, the water holding capacity of the coating polymers used can change the product moisture. According to the wall material selected, the technological and rheological characteristics of the fortified bread can be modified. Hadidi et al. [60] reported that the highest moisture content was found in the bread supplemented with encapsulated *L. acidophilus* in alginate/fish gelatin at 3%, followed by alginate/fish gelatin at 1.5%.

Similarly, Ghasemi et al. [1] showed that bread with tragacanth gum microcapsules (used for the encapsulation of *L. acidophilus* and *L. plantarum*) had a higher moisture content than other gluten-free samples, ascribing the behavior to the ability of gums in binding and retaining water. In contrast, the moisture content of the bread with encapsulated L. rhamnosus did not have any significant effect on the moisture content, regardless of the baking condition and wall material [14]. The addition of alginate-encapsulated *L. acidophilus* had no significant effect on the oven spring—that is, the growth of the bread during its initial baking phase where the loaf is growing before the crust hardens—and the specific volume of the bread samples [60]. However, alginate and fish gelatin microcapsules significantly affected these two parameters. Indeed, the rise in the oven spring and specific volume values of the bread could be attributed to an increased instability of the gluten–starch network, increased dough strength, and increased gas retention capacity [53,61]. In contrast, Ezekiel et al. [14] reported that encapsulated probiotics in the bread and baking condition did not cause a significant alteration in terms of specific volume. In addition, gluten-free bread enriched with probiotic bacteria encapsulated with tragacanth gum had a higher specific volume and oven spring than encapsulation with sago starch and control bread, as reported by Ghasemi et al. [1]. An effect of the tragacanth capsules is probably due to the hydroxyl groups of the gum, which increased the water absorption and thus the volume [62]. Due to hydrocolloids, the gluten-free dough viscosity is raised, likely improving the retention of gas bubbles during mixing and fermentation, and the water vapor during baking is also raised, thus creating a porous structure with a high specific volume. The use of hydrocolloids for the encapsulation of probiotic cells also leads to the modification of texture. The encapsulation of *L. plantarum* and *L. acidophilous* with tragacanth gum and sago starch significantly lowered the hardness of the gluten-free bread compared with the control [1]. Moreover, tragacanth gum reduced the hardness and staling of bread more effectively than sago starch. Indeed, gums have a high binding capacity with water and prevent moisture loss during the baking process. In addition, the reaction between gum and starch could postpone starch retrogradation and delay the crumb-stalling effect. Similarly, Hadidi et al. [60] found an inferior crumb hardness and stalling rate in bread with encapsulated L. acidophilous compared with the control sample. Moreover, microcapsules with alginate and an increased amount of fish gelatin [0.5, 1.0, 1.5%] showed increased power in lowering the hardness and stalling rate. This result was probably related to more moisture content and a larger specific volume of bread caused by the increment of fish gelatin.

### 2.6. Enzymes

Breadmaking involves the use of enzymes deriving from three sources: endogenous enzymes naturally occurring in flour, enzymes related to the metabolic activity of yeasts, and other dominant microorganisms and exogenous enzymes, which are intentionally incorporated in the dough [63]. The extensive use of exogenous enzymes has gained much importance due to restrictions in the addition of synthetic additives. Indeed, enzymes can act as flour standardizer, dough rheology modifiers, and improvers of textural properties and can be incorporated individually or in combination, encapsulated or not. Among the class of enzymes implied in baking, the endo-acting α-amylases are commonly used for their role in retarding staling and improving and/or standardizing flour (Table 6). The anti-staling efficiency relies upon the ability to limit the formation and the strength of the amylopectin network and act for water immobilization. In the work of Haghighat-Kharazi et al. [13] α-amylase derived from Bacillus licheniformis was encapsulated in beeswax to assess the effect in gluten-free bread, for which the staling represents a major issue. The catalytic efficiency was about two-fold lower and the thermal and storage stability were higher compared with the free-added enzyme. Furthermore, gluten-free bread with encapsulated α-amylase resulted in having a lower hardness and chewiness and higher sensory acceptability. Further research led to the investigation of the addition of maltogenic amylase derived from Bacillus stearothermophilus, as this exo-acting enzyme was discovered to be one of the most effective anti-staling amylases [13]. The authors tested the encapsulation of maltogenic amylase in low- and high-dextrose equivalent maltodextrins. Once incorporated into gluten-free bread, the different formulations showed no significant differences in moisture content and firmness parameters. Moreover, the higher softness of the crumb concerning the control samples was observed. In addition, bread with the low-dextrose equivalent maltodextrin had a higher uniformity of gas cells and lower weight loss. Afterwards, the use of beeswax, alone or with maltodextrin as the wall materials, for encapsulating maltogenic amylase in gluten-free bread was investigated [64,65]. Gluten-free bread with maltogenic amylase encapsulated with both beeswax alone or combined with maltodextrin showed an overall higher quality and sensory acceptability compared with the control samples. Furthermore, bread with the encapsulated enzyme was compared with the non-encapsulated one. The results showed a higher softness of the crumb for the bread with unencapsulated enzyme, but a higher sensorial acceptability for the one with microcapsules for up to 4 days of storage.

Glucose oxidase is another enzyme widely used in baking products for its antimicrobial properties because of its ability to remove glucose and oxygen residuals, producing hydrogen peroxide. However, the application in bread presents some drawbacks due to its low stability in dough and rapid oxidation. In this context, Zhang et al. [66] tested the use of encapsulation as a tool to protect glucose oxidase when added to Chinese steamed bread. The results showed that bread with microencapsulated enzyme had a slower action in catalyzing the oxidation of dough, resulting in higher extensibility and wet gluten content of the dough, better texture properties, and higher sensory acceptability.

### 2.7. Other Bioactive Compounds

Within the frame of improving the sensory attributes of cheese bread, Silva et al. [68] proposed the addition of microencapsulated Swiss cheese bioaroma in cheese bread. The spray-dried microcapsules were obtained using maltodextrin and corn starch in a ratio of 1:1, and 4 formulations of bread were prepared with 0.0, 2.2, 4.4, and 6.6% of the encapsulated cheese bioaroma. Regarding the structural and technical quality parameters, in treated samples the texture improved with increasing concentrations of the bioaroma because of the presence of modified starch and maltodextrin. Furthermore, the sensory acceptability indicated that the bread with 6.6% of encapsulated bioaroma was the preferred formulation, thus suggesting that the additive used gives a distinctive aroma to the cheese bread without (undesirably) affecting the structural and other sensory attributes of the product.

## 3. Challenge and Prospects

Although many studies have successfully fortified bread with encapsulated bioactive compounds and achieved good experimental results, many challenges need to be addressed. Encapsulated bioactive compound inclusion is promising because of the prospect of improving product functionality without compromising bread quality or safety; however, there is still much research to be conducted to enable broad-spectrum applications for food fortification. The microencapsulation process increases the cost of the final product, which is the main limitation of industrial production. Therefore, the investigation of new technologies and low-cost shell materials is of interest. In addition, as highlighted in this review, much research did not investigate the effect of the microcapsules’ inclusion on the quality of the final product, limiting the understanding of the potential of this application. Many researchers have focused their efforts on understanding the interaction between bioactive compounds and coating polymers. The same should be carried out to assess the interaction of the obtained microcapsule with the final product as the effect of the inclusion is specific to the different products. As has emerged from the current review carried out, the selection of suitable coating polymers and specific microencapsulation conditions can lead to an improvement in the rheological and sensory characteristics of the bread. Taking advantage, as an example, of the water-holding capacity of the shell materials, it is possible to increase the moisture in functional samples, reduce their hardness and, at the same time, increase their shelf-life.

## 4. Conclusions

Bread is one of the most broadly consumed food products, and it is also characterized by a high acceptability. Given these features, it is considered a valuable matrix for the inclusion of bioactive compounds such as vitamins, minerals, phenolic extract, source of PUFA, probiotics, and enzymes. The inclusion of the free form of the bioactive compounds in this kind of food is challenging because of their loss during backing and storage. Considering the literature reviewed here, it is possible to conclude that the encapsulation of the bioactive compounds is an efficient tool to overcome the cited drawbacks. In addition, the bioactive compounds encapsulation exerts positive impacts on the physiochemical and rheological bread properties. As an example, it emerged that the encapsulation increases the bread shelf-life as well as the sensory characteristics considering the fortification with PUFA sources. A priority is the selection of the wall materials: polymers with different hydrophilicity have different water-holding capacities, which affects the bread moisture and hardness. Altogether, it is possible to conclude that the encapsulation of bioactive compounds for bread fortification is a promising technology. However, to be able to use the encapsulated bioactive compounds in commercial bread, other challenges must be dealt with: primarily, the economic aspects linked to the production of the microcapsules; and secondly, the scalability of the process. To date, only a single study has evaluated the production of bread fortified with encapsulated bioactive compounds at a commercial bakery scale.

## Figures and Tables

**Table 1 foods-12-00096-t001:** Advantages and disadvantages of microencapsulation techniques.

Technique	Size	Advantages	Disadvantages
Spray drying	1 μm–100 μm	Rapid process, cost-effective, simple continuous process, reproducible, high productivity, and easy scale up	Higher temperature, a broad range of size distributions, and the range of polymers that can be used is rather limited
Freeze drying	100 nm–5 mm	Simple process, low temperature, and absence of air	Greater production time, a broad range of size distribution, and capital costs
Coacervation	1 μm–5 mm	Simple process, low temperature, and low evaporation loss	Expensive and complex, difficult scale-up, batch process, and additional drying process is required
Fluidized bed coating	10 μm–20 mm	Economical, fast, high production, and use of different coating material	Higher temperature, relatively difficult to master a longer duration
Extrusion	1 mm–5 mm	Lower temperature, simple, and low cost	Unable to form microcapsules in viscous coating material, high cost, and slow technique

**Table 2 foods-12-00096-t002:** Encapsulation of vitamins and minerals for bread fortification.

Core	Wall	Technique	Keys Finding(s) and Recommendation	Ref.
Vitamin D	Egg white proteins	Ultrasonication	↑ Resistance of vitamins to light and heat ↑ protection of vitamins from mechanical stress ↑ recovery rate 81.3%	[24]
Vitamin D	Amaranth protein isolates and lactoferrin	n.d.	↑ Recovery rate 88.6% ↑ absorption of vitamins in the intestine	[25]
L-5-MTHF ascorbate	Modified starch	Spray drying	↑ Stability of vitamins during the bread baking process ↑ storage stability of bread	[26]
L-5-MTHF	Skim milk powder	Spray drying	↑ Stability of L-5-MTHF	[27]
Folic acid	n.d. (commercial powder)	n.d. (commercial powder)	↓ Resistance to thermal treatment	[28]
Iron	Modified starch	Spray drying	↑ Bioaccessibility for conventional bread-making process	[11]

L-5-methyltetrahydrofolic acid (L-5-MTHF): ↑ higher quantity/effect; ↓ lower quantity/effect; n.d.: no data.

**Table 4 foods-12-00096-t004:** Encapsulation of phenolic extracts for bread fortification.

Core	Wall	Technique	Keys Finding(s) and Recommendation					Ref.
			EE	Phenolic Content and Antioxidant Activity	Physicochemical Characteristics	Texture	Sensory Analysis	
Garcinia cowa fruit extract	WPI or MD or WPI + MD	Freeze drying	Above 90%	↑ HCA (171% MD, 172% WPI + MD and 185 % WPI)	≈ Moisture ↓ volume for MCs bread ↑ volume for WPI (among experimental bread)	↑ Crumb hardness for MCs bread ↓ crumb hardness for WPI (among experimental bread)	↑ Acceptability for WPI	[42]
Garcinia cowa fruit extract	WPI or MD or WPI + MD	Spray drying	↑ MD	↑ HCA (86% for MD)	≈ Moisture ↓WPI, MD, WPI + MD hide the extract color	↑ Crumb softness for MD (among experimental bread)	↑ Acceptability for MD	[10]
Green tea extract	MD or β -CD or MD-βCD	Freeze drying Spray drying	MD ↑EE for both the techniques	↑ PC for freeze-dried MD	↑ Moisture for MCs bread ≈ volume for MCs bread ↑ dark color	≈ Hardness	↓ Sensory quality characteristics	[43]
Saskatoon berry fruit extract	MD or I	Freeze drying	n.d.	↑ AA and PC for MD	↑ Dark color for MCs bread	n.d.	↑ Overall acceptability for MCs bread with 3% of encapsulated extract	[44]
Saskatoon berry fruit extract	MD or I	Freeze drying	n.d.	↑ AA and PC for MCs bread ≈ AA and PC among the experimental bread	↑ Dark color for MCs bread	n.d.	↑ Overall acceptability for bread with 3% of MCs	[45]
Soybeans, onion, young hawthorn extracts	MD or I	n.d.	n.d.	↓ PC for hawthorn extract ≈ AA for experimental bread	↑ Yield and ↓ volume for experimental bread	n.d.	n.d.	[46]

Microcapsules [MCs]; encapsulation efficiency [EE]; inulin [I]; maltodextrin [MD]; β-cyclodextrin [β -CD]; whey protein isolate [WPI]; phenolic content PC]; antioxidant activity [AA] ≈ no effect/no significative effect; ↑ higher quantity/effect; ↓ lower quantity/effect; n.d.: no data.

**Table 5 foods-12-00096-t005:** Encapsulation of probiotics for bread fortification.

Core	Wall	Technique	Keys Finding(s) and Recommendation					Ref.
			EE	Gastro-Intestinal Resistance	Survivability in Bread	Physicochemical Characteristics	Texture	
*L. rhamnosus* LGG	Single-layer [Sl]: Na-Al Multiple-layer [Ml]: Na-Al + C, Na-Al + CS, Na-Al + HM-RS, Na-Al + CS + C, and Na-Al + HM-RS + C	Extrusion	98.1–99.88% ↑ EE for Ml	↑ For Ml wall	↑ For Ml	≈ Dough weight ≈ volume of bread ≈ specific volume	n.d.	[14]
*L. acidophilus * and *L. plantarum*	TG or SS or TG + SS	Emulsion	n.d.	n.d.	n.d.	≈ pH of bread Specific volume with TG Mcs ↑Oven spring for TG Mcs ↑Moisture for bread for TG Mcs	↓ Hardness for TG Mcs bread	[1]
*L. acidophilus*	Na-Al or FG	Emulsion	↑ For Na-Al + FG ↓ For Na-Al alone	n.d.	↑ For Na-Al + FG in bread and after 7 days of storage	↑ Moisture in 7 days of storage for FG capsules ↑Volume	↓ Hardness in 7 days of storage for FG	[60]
*L.sporogenes*	MCC + Na-Al or MCC + XG	Fluidized bed method	↑ For MCC + I + Na-Al	↑ For XG 1.5% bread	↑ for GE [1.5%]	n.d.	n.d.	[54]
*L. acidophilus *	Na-Al or C or XG or GE	Fluidized bed method	↑EE for XG 1% as first layer coating	↑ For 1% Na-Al or XG	n.d.	n.d.	n.d.	[55]
*L. casei* and *L. acidophilus*	Ca-Al + HMRS Ca-Al + HMRS + XG	Extrusion	n.d.	n.d.	↑ Ca-Al + HMRS + XG ↑ in Hamburger bun than Pan bread	n.d.	n.d.	[58]
*B. lactis*	Na-Al +Hpc + MCC	n.d.	n.d.	↑ For encapsulated bacteria compared free	↑ For encapsulated than free bacteria	n.d.	n.d.	[56]

Microcapsules [MCs]; encapsulation efficiency [EE]; sodium alginate [Na-Al]; chitosan [C]; cassava starch [CS]; high maize resistance starch [HM-RS]; tragacanth gum [TG]; sago starch [SS]; fish gelatin [FG]; microcrystalline cellulose [MCC]; inulin [I]; xanthan gum [XG]; calcium alginate [Ca-Al]; gellan [GE]; hydroxypropyl cellulose [Hpc] ≈ no effect/no significative effect; ↑ higher quantity/effect; ↓ lower quantity/effect. n.d.: no data.

**Table 6 foods-12-00096-t006:** Encapsulation of enzymes for bread fortification.

Core	Wall	Technique	Keys Finding(s) and Recommendation						Ref.
			EE	Catalytic Efficiency	Thermal and Storage Stability	Physicochemical Characteristics	Texture	Sensory Analysis	
α-amylase	BW	Emulsion-congealing technique	40%	↓ For encapsulated enzyme	↑ For encapsulated enzyme	n.d.	↓ Hardness and chewiness	↑ Overall acceptability for encapsulated enzyme (vs. free enzyme and control)	[66]
Maltogenic amylase	MD with 2 DEs: LMD and HMD	Emulsion-congealing technique	↑ For LMD (93% vs. 68% of HMD)	n.d.	n.d.	≈ Moisture and firmness ↓ weight loss for LMD	↑ Softness of the crumb for LMD	≈ Overall acceptability	[13]
Maltogenic amylase	MD + BW	Emulsion-congealing technique	79%	n.d.	n.d.	≈ Crumb/Crust ratio (vs. free enzyme and control) ↑ crust dark color (vs. control)	↓ Hardness and gumminess (vs. free enzyme and control) ↓ chewiness (vs. control)	↑ Overall acceptability (vs. control)	[64]
Maltogenic amylase	BW	Emulsion-congealing technique	42%	n.d.	n.d.	↓ Crumb/crust ratio (vs. control) ↑ crust dark color (vs. free enzyme and control)	↑ Softness of the crumb (vs. free enzyme)	↑ Overall acceptability (vs. free enzyme and control)	[65]
Glucose oxidase	Na-Al + C	Emulsification/internal gelation	n.d.	↓ Oxidation speed of encapsulated enzyme	n.d.	↑ Wet gluten content (vs. control) ≈ wet gluten content (vs. free enzyme) ↑ extensibility and specific volume	↓ Crumb hardness	↑ Overall acceptability	[67]

Beeswax [BW]; low-high dextrose equivalent [L-HDE]; maltodextrin [MD]; sodium alginate [Na-Al]; chitosan [C] ≈ no effect/no significative effect; ↑ higher quantity/effect; ↓ lower quantity/effect. n.d.: no data.

## Data Availability

Not applicable.

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
