# Peer review of "Current Advantages in the Application of Microencapsulation in Functional Bread Development"

_foods, 2022, doi:10.3390/foods12010096_

Round 1

Reviewer 1 Report

After carefully reviewing of the manuscript “Current advantages in the application of microencapsulation in functional bread development”, I can feel the hardworking of the author to summarize so many literatures. But I can’t get some clear and useful information as a reader after the first reading of the manuscript. Here are some comments to improve the manuscript:

1.     The most important thing for a review is to summarize the previous literature to provide some clear hint for the future academics. The authors presented lots of previous studies and listed results in several aspects, which makes the reader can’t focus on the most important issue. And what’s the most important issue the author wanted to address? The effect of the encapsulation on the bioactive? The effect of encapsulated bioactive on the bread property? Or the effect of the wall material (hydrocolloids) on the bioactive or bread?

So based on the numerous information, the authors should re-evaluate their main focus and what they want to convey to readers to make then benefit from your manuscript.

2.     Secondly, the authors should raise some point and discus based on the previous research. They pointed out some question in the conclusion, but they didn’t discus it properly. I think they can add a “challenge and prospect” section.

3.     Table 5 is not necessary to be presented because the information is all about the encapsulation effect of the β-carotene, not the effect on the bread. The authors should focus on the “application of microencapsulation in functional bread development”, not the encapsulation.

4.     The similar question to 2.6.1, first it is known that the encapsulation would enhance the GIT tolerance of the probiotic, secondly, the review should focus on bread property. The whole part of 2.6.1 is not necessary in this review.

5.     There are some information missing for the references, please check. Such as reference 30 and 34.

Reviewer 2 Report

Dear Authors,

During reviewing process I have noticed a few problems listed below.

1.   1. Minor language correction is needed, e.g. „Microencapsulation process and tecniques” – should be techniques.

2.   2.  Part “Bread rheological and textural properties” – The rheological parameters are thixotropy or yield stress (it is changing the viscosity under the influence of stress/force). Could you please change the name of this part. “The texture properties” appears to be sufficient.

3.   3.  Part „2.4.3 Bread technological and rheological parameters” – the use of the word rheological is inappropriate (please remove this word).

4.    4. Please specify the name of presented subchapter “2.6.2 Bread characteristics” – bread containing probiotics?

Round 2

Reviewer 1 Report

The revised version is acceptable.